# Resistance to Antiangiogenic Therapy in Hepatocellular Carcinoma: From Molecular Mechanisms to Clinical Impact

**DOI:** 10.3390/cancers14246245

**Published:** 2022-12-18

**Authors:** Piera Federico, Emilio Francesco Giunta, Andrea Tufo, Francesco Tovoli, Angelica Petrillo, Bruno Daniele

**Affiliations:** 1Medical Oncology Unit, Ospedale del Mare, 80147 Naples, Italy; 2Department of Precision Medicine, School of Medicine, University of Study of Campania “L. Vanvitelli”, 80131 Naples, Italy; 3Surgical Unit, Ospedale del Mare, 80147 Napoli, Italy; 4Department of Medical and Surgical Sciences, University of Bologna, 40126 Bologna, Italy; 5Division of Internal Medicine, Hepatobiliary and Immunoallergic Diseases, IRCCS Azienda Ospedaliero-Universitaria di Bologna, 40138 Bologna, Italy

**Keywords:** hepatocellular carcinoma, primary resistance, secondary resistance, immune checkpoint inhibitors, molecular targeted drugs, Wnt/β-catenin, vessel normalization

## Abstract

**Simple Summary:**

Resistance to antiangiogenic therapy represents a challenge in the therapeutic approach for hepatocellular carcinoma (HCC). The advent of immune checkpoint inhibitors (ICIs) seems to be a solution to this issue; however, other gaps have emerged since they have only been shown to provide benefit in approximately 20% of patients. Combination strategies have introduced in clinical development to overcome this issue. This review deals with the resistance to antiangiogenic drugs by focusing on two protagonists, the tumor microenvironment and vascular normalization, to offer possible overcoming strategies.

**Abstract:**

Antiangiogenic drugs were the only mainstay of advanced hepatocellular carcinoma (HCC) treatment from 2007 to 2017. However, primary or secondary resistance hampered their efficacy. Primary resistance could be due to different molecular and/or genetic characteristics of HCC and their knowledge would clarify the optimal treatment approach in each patient. Several molecular mechanisms responsible for secondary resistance have been discovered over the last few years; they represent potential targets for new specific drugs. In this light, the advent of checkpoint inhibitors (ICIs) has been a new opportunity; however, their use has highlighted other issues: the vascular normalization compared to a vessel pruning to promote the delivery of an active cancer immunotherapy and the development of resistance to immunotherapy which leads to a better selection of patients as candidates for ICIs. Nevertheless, the combination of antiangiogenic therapy plus ICIs represents an intriguing approach with high potential to improve the survival of these patients. Waiting for results from ongoing clinical trials, this review depicts the current knowledge about the resistance to antiangiogenic drugs in HCC. It could also provide updated information to clinicians focusing on the most effective combinations or sequential approaches in this regard, based on molecular mechanisms.

## 1. Introduction

Hepatocellular carcinoma (HCC) is one of the most frequent malignancies in the world [1]. The main risk factors for its onset are chronic hepatitis B or C virus infection, alcohol abuse and metabolic syndrome related to diabetes and obesity [1,2].

The peculiar vascularization of the liver—perfused by two afferent circulations (portal vein and hepatic artery)—underlines the importance of angiogenesis in HCC. This was the prerequisite for the development of antiangiogenic therapy, both in the early/intermediate stages and in the late stages of the disease [3]. Particularly in advanced disease, sorafenib represented the only effective first-line therapy for a decade [4]. Then, lenvatinib proved the non-inferiority compared with sorafenib [5]. Additionally, regorafenib, cabozantinib and ramucirumab received approval as second-line treatments after sorafenib [6,7,8]. Nevertheless, although these drugs have been shown to improve clinical outcomes, the median overall survival (OS) remains poor and drug resistance might be considered the principal cause of treatment failure during targeted therapies. Based on this background, the main goal of HCC targeted therapy has shifted from the concept of “tumor starving” to the need for delivering the drugs through “vascular-normalization” [9,10]. 

In this context, immunotherapy -alone or in several combinations with antiangiogenic agents- has emerged as a major therapeutic modality in the last decades in oncology. Regarding HCC, the IMbrave150 trial showed that the combination of atezolizumab and bevacizumab improves OS in the first-line treatment for BCLC C and recurrent BCLC B patients [11]. Interestingly, as exploratory analysis, the trial focused also on the tumor microenvironment (TME), on the changes induced in it by vascular endothelial growth factor (VEGF) and targeted therapy against VEGF receptor (VEGFR), able to convert a “cold” tumor, such as HCC, into an “immunologically hot” one [12].

Thus, if immunotherapy might be considered a way to overcome the resistance induced by targeted therapy alone, it must be said that—if administered as a single agent—it fails to be responsive in a major proportion of patients with HCC: in fact, 40% of patients are primarily resistant to immune checkpoint inhibitors (ICIs) [13]. The mechanisms of resistance to ICIs include different aspects: failure in antigen presentation, heterogeneity in the TME, alterations in the regulatory molecules of ICIs and influence of immune-suppressive cells [13].

Based on this background, we provide an overview of the mechanisms of resistance to targeted therapy and immunotherapy in HCC, by focusing on the role of angiogenesis through the modulation of both the tumor vasculature and the TME. 

## 2. Angiogenesis in Hepatocellular Carcinoma: From the Anatomical Point of View to Clinical Relevance

The human liver is organized in lobules, roughly hexagonal in shape with a repetitive structure, the portal triad (branch of portal vein, branch of the hepatic artery, bile duct) that sits at each corner of the hexagon. The hepatic artery and the portal vein drain into capillary-like structures called sinusoids, which exchange materials directly with the hepatocytes via the fenestrated endothelium [14,15].

The bile and blood circulations work in opposite directions: a centrifugal direction and a centripetal flow, respectively; this feature emphasizes the known peculiarity of the normal liver about the vascularization, on the one hand being interposed between two venous systems and on the other, having an arterial supply afferent to a venous system, the sinusoids, shared with the venous afference.

Liver tumors display a hypervascular nature; four mechanisms have been described to acquire new blood vessels: co-option, sprouting, vasculogenesis and intussusception [16]. Whichever mechanism is used by the tumor cell among those mentioned, hypoxia remains the triggering factor of tumor angiogenesis. In fact, although HCC is a hypervascular type of cancer, it is also characterized by hypoxia: HCC arises from a chronic liver injury inducing fibrogenesis, ultimately resulting in cirrhosis. This latter destroys the normal hepatic vascular system leading to hypoxia that stimulates tumor angiogenesis. Hypoxia-induced factor-1 (HIF1) is the main transcription factor activated by hypoxia. It acts in the nucleus on hypoxia-responsive elements which induce the expression of genes whose products are proangiogenic factors: insulin-like growth factor 2 (IGF-2), VEGF, angiopoietin-2 (Ang2), platelet-derived growth factor (PDGF)- β and others [17,18,19,20]. 

VEGF plays a critical role in mediating angiogenetic effects as it induces peculiar vascular changes mainly through VEGF/VEGFR2 interaction [10]: arterialization of blood supply and sinusoidal capillarization; hypovascular areas and severe hypoxia and/or necrosis; an abnormal microenvironment that selects increasingly aggressive tumors [21,22].

Another aspect that should be considered is the active role of the hepatitis B virus (HBV) and hepatitis C virus (HCV) in carcinogenesis and tumor progression in HCC [23]. In detail, HBV DNA integrates into host chromosomes, causing gene de-regulation in preferred target regions, but also on the epigenetic level through DNA methylation and/or histone modifications: all these alterations cause impairment of the main pathways involved in HCC development, including Wnt/β- and angiogenic ones [24,25]. Concerning HCV, a similar influence on molecular pathways has been highlighted [26,27].

In conclusion, VEGF and its receptors are critical protagonists in the angiogenesis of HCC; their expression is high in HCC cell lines and tissues as well as in blood circulation, especially for patients with more aggressive disease [28]. Thus, antiangiogenic drugs represent a cornerstone in the treatment of HCC.

## 3. Resistance to Antiangiogenic Drugs

As HCCs typically have arterial hypervascularity, the traditional treatment approach using anti-angiogenic therapies has set itself the goal of starving the tumor by depriving oxygen and nutrients. Systemic therapies with small molecules act on angiogenesis by blocking multiple tyrosine kinases [29]. However, even if a detailed description of all the approved drugs in this setting is not the aim of this review, resistance to antineoplastic drugs is a critical issue in clinical practice, because it is the main cause of therapeutic failure against cancer [30].

In this regard, we can divide resistance into two main classes, bearing in mind that this distinction is more theoretical than practical: 

Primary (intrinsic) resistance, which means that drugs do not work in that specific patient either for some preexistent characteristics of the patient itself (i.e., pharmacokinetic alterations, genetic predisposition etc.) or because of intrinsic “indifference” of the tumor (i.e., pre-existent modifications of the molecular target, different cellular metabolism).

Secondary (extrinsic) resistance, which implies a first phase of sensitivity to antineoplastic drugs followed by the de novo onset of both pharmacokinetic and/or pharmacodynamic changes that led to specific unresponsiveness, resulting in tumor progression [31,32].

From a clinical point of view, there is no unique time value threshold for separating primary from secondary resistance, but it is accepted that a rapidly progressive disease in the very first months of treatment is attributable to primary resistance, as well as to the secondary one if it comes later [33].

### 3.1. Sorafenib

Among the antiangiogenic drugs used for treating advanced and metastatic HCC, sorafenib is the most studied drug.

Sorafenib has been used as a single-agent drug in treatment-naïve advanced HCC since 2007 [34]. The two phase III clinical trials which led to its approval showed that nearly one-half of patients progressed before 4–5 months from starting therapy, highlighting the weight of intrinsic resistance in this therapeutic scenario [4,35].

Several molecular mechanisms of primary resistance to sorafenib have been discovered to date. First of all, drug efflux and uptake can negatively affect sorafenib efficacy by reducing its intracellular concentration [36]. This mechanism, OCT1 (Organic Cation Transporter 1), encoded by the SLC22A1 gene, is responsible for sorafenib uptake, and its down-regulation is associated with shorter OS in HCC patients treated with this specific drug [37]; P-glycoprotein, also known as MDR-1 (multidrug resistance protein 1) or ABCB1 (ATP-binding cassette sub-family B member 1), causes expulsion of sorafenib from the intracellular space [38]. Regarding target alterations, some polymorphisms of the VEGFR2 gene have been correlated to reduced survival in HCC patients treated with sorafenib [39], whilst the increased activity of survival pathways such as MAPK/ERK, PI3K/AKT or Hedgehog has been associated with lower sensitivity to sorafenib in in vitro HCC models (36).

Acquired resistance to sorafenib is mainly driven by alterations in angiogenetic mechanisms: (a)Disruption of tumor vessels could select resistant cells overexpressing HIF-1α, which is a transcription factor regulating angiogenesis and, definitely, tumor progression [40];(b)Other mechanisms of acquired resistance could be the “co-option of liver vessels”, that is recruitment of pre-existing liver vessels by HCC without resorting to neoangiogenesis [41]; abnormal activation of PI3K/Akt and JAK-STAT pathways and fibroblast growth factor (FGF) signaling pathways [42]; elevated expression of MET [43]; genome instability and epigenetic regulation [43].

Then, the immune microenvironment and inflammation are also involved in sorafenib resistance. It must be remembered that HCC arises in the context of chronic inflammation in most cases, and microenvironmental inflammatory processes are responsible for tumor onset and progression [44]. Increased TNF-α and TGF-β levels have been associated with poor prognosis in patients treated with sorafenib [45,46], whilst expression of CCL2 and CCL17 by tumor-associated neutrophils could be responsible for sorafenib resistance [47]. Additionally, patients with high levels of intratumoral CD8+ cytotoxic T lymphocytes expressing programmed death receptor-1 (PD-1) showed poor prognosis when treated with sorafenib [48].

All these data support the hypothesis of an impact on sorafenib efficacy by different settings of TME. However, to date, no predictive factor has been identified for use in clinical practice, even if several tissue and blood biomarkers are under investigation [49].

### 3.2. Lenvatinib

Lenvatinib has shown to be non-inferior to sorafenib in prolonging OS when used as first-line therapy in the REFLECT trial, even if some differences in clinical activity should be pointed out: a longer progression-free survival (PFS) and greater overall response rate (ORR) in both investigator and independent review. This data highlighted an intrinsic difference between the two molecules, especially concerning primary resistance mechanisms [5].

The analysis of serum biomarkers in patients enrolled in this trial has detected the potential negative prognostic role of baseline VEGF, ANG2 and FGF21, regardless of the drug used [50], whilst elevation of FGF23 levels, as previously reported in thyroid cancer patients responding to lenvatinib therapy [51], was observed when complete or partial responses were reached in HCC patients treated with lenvatinib. Upregulation of the HGF/c-MET axis has determined acquired resistance to lenvatinib in a preclinical model of HCC [52], suggesting its potential role as a therapeutic target to overcome lenvatinib resistance, particularly in light of the high percentage of HCC patients overexpressing these two molecules [53]. Expression levels of PAK1, a serine/threonine protein kinase involved in several cancer signaling pathways including MET axis [54,55], could be responsible for lenvatinib resistance [56].

### 3.3. Regorafenib and Cabozantinib

Despite disease control in two-thirds of sorafenib-resistant HCC patients on second-line treatment with regorafenib, half of the patients will experience progression of disease within 3 months from starting therapy [6]. Biomarker analysis in patients enrolled in the RESORCE trial has shown that plasma levels of angiopoietin 1, which promotes tumor proliferation and angiogenesis, are prognostic in patients treated with regorafenib but not in the overall population [57]. Molecular resistance to regorafenib may be explained by changes in the TME, namely fibrosis, hypoxia and inflammation [58]. 

However, from the molecular point of view, few mechanisms have been described to date. Sphingosine kinase 2 (SphK2) and sphingosine-1-phosphate (S1P) determine regorafenib resistance through NF-kB activation in preclinical models of HCC [59], whilst low levels of secreted glycoprotein ADAMTSL5 restores regorafenib sensitivity [60]. Intracellular mitogenic and antiapoptotic processes could be also involved in regorafenib resistance [61,62].

Regarding cabozantinib, mechanisms of resistance in HCC preclinical models or patients have not yet been reported. However, given the profile of target inhibition, cross-resistance of cabozantinib with other antiangiogenetic drugs is expected with the exception of the hyper-expression of AXL and MET receptors, which are specifically inhibited by this drug [63].

### 3.4. Anti-VEGF/VEGFR Monoclonal Antibodies

Bevacizumab is currently used as a first-line therapy in combination with atezolizumab; thus, the discrimination between resistance to anti-VEGF or anti-PD-L1 (or both) is difficult in case of disease progression. Biomarkers of primary and acquired resistance are essential to identify patients in which the solely VEGF blockade allows tumor to carry on angiogenesis through different pathways (i.e., PDGFR, fibroblast growth factor receptor (FGFR), MET, etc.).

Ramucirumab efficacy as a second-line therapy is limited to patients with high plasmatic levels of AFP (>400 ng/mL), as mentioned before. AFP seems to be related to angiogenesis in specific subtypes of HCC [64]. To date, the clinical role of AFP is not completely clear: it has shown to be a prognostic factor in both early and advanced stages [65]; however, its predictive role has been validated only for ramucirumab treatment. A retrospective analysis of CELESTIAL trial has shown the potential role of high baseline levels (>400 ng/mL) of AFP in predicting longer PFS for patients treated with cabozantinib [66], whilst subgroup analysis of IMbrave150 has shown a better OS of the doublet over sorafenib only in patients with low baseline levels (<400 ng/mL) of AFP [11].

To date, specific mechanisms of molecular resistance to anti-VEGF(R) monoclonal antibodies in HCC are not known, even if seems reasonable that they do not differ from other cancer types [67] (Figure 1).

## 4. Overcoming Resistance: From the Concept of “Starve the Tumor” to the “Vessel Normalization”

The concept of the normalization of vascularity is an emerging alternative point of view if compared to the conventional therapeutic effect of starving the tumor; the consequent increased perfusion and oxygenation are able to enhance the delivery and effectiveness of therapies to the tumor. The goal of the anti-angiogenic therapy, in this perspective, is not only to inhibit the vascular system but also to modify the TME, which will be characterized by less hypoxia, less vascular leak, an increase in the number of pericytes along with an increase in CD8 + T cell infiltration and a decrease in the neutrophil/lymphocyte ratio [68]. Such changes lead to a potential increase in the delivery of other cancer-directed therapies including ICIs and could be used to improve their effectiveness (Figure 2).

Normalizing the vasculature means carrying out a selective pruning of immature tumor vessels [69]. An adequate dose of antiangiogenic agent leads to an appropriate regression of the vasculature, which assumes the structure and phenotype similar to those of normal, non-tumor vessels. By “normalization window” we mean precisely the window of time from the beginning of antiangiogenic therapy in which the adequate dose of the drug is reached leading to the correct pruning of vessels, neither excessive nor reduced, and therefore to better oxygenation of cancer cells with a greater vulnerability to cytotoxic therapies [70,71]. High doses of antiangiogenic therapy can result, in fact, in a reduced normalization window causing the excessive pruning of vessels, hypoxia, acidosis in TME, increased deposition of extracellular matrix and infiltration of immunosuppressive and pro-tumor cells such as monocytes or myeloid-derived suppressor cells (MDSCs) [72]. Low doses of antiangiogenic treatment, on the other hand, could determine a prolonged normalization giving the tumor an aggressive phenotype [73] (Figure 2). 

In this context of defining the normalization window in experimental models of different tumors, it was found that the signal mediated by the Ang1/Ang2-Tie2 axis is important for the recruitment of pericytes for tumor vessels and that the overexpression of Ang2 may negate the benefit of anti-VEGF therapy by abrogating Ang1-mediated vessel normalization [74]. As discussed, this means that the concomitant blockade of Ang2 and VEGF extends both the normalization window and survival compared to blocking only one pathway and Ang2 could be a predictive/prognostic marker of treatment efficacy [75]. Therefore, a greater understanding of the interaction between the VEGF signal and Ang2 can offer new therapeutic strategies.

The normalization window for HCC is not known, unlike other carcinoma models such as, for example, melanoma, breast cancer and ovarian cancer [68]. However, this alternative view has highlighted other functions of the VEGF signaling pathway in the TME. In particular, achieving normalization of the vessels means ensuring an infiltration of immune cells and reprogramming the TME from an immunosuppressive into immune-supportive microenvironment [76]. VEGF, in fact, suppresses the antitumor immune response and promotes the accumulation of tumor-associated macrophages (TAM), regulatory T cells (Treg cell) and MDSCs in tumor tissue and secondary lymphoid organs, resulting in a pro-tumorigenic microenvironment [77]. Among immune cells, Tregs have been identified as a major source of VEGF in TME. Tregs modify the phenotype of TAMs, which express B7H in favor of the pro-tumor M2 phenotype [78]. VEGF induces the expression of PD-1 on CD8 + T and Treg cells and hypoxia through HIF1a directly causes the upregulation of PDL1 and CTLA4 on MDSC, TAM, dendritic cells (DC) and tumoral cells. In this way, VEGF controls the binding between PD-1 and its ligand (PD-L1) and the immune response of CD8 + T cells is negatively regulated. Following exposure to VEGF, DC also contribute to antitumor immune response suppression as they lose their ability to present antigens and to recruit CTLs.

Overall, immature Treg, TAM, MDSC and DC cells suppress the activity of CTLs, thus resulting in a pro-tumorigenic microenvironment. Anti-VEGF therapies, working for a normalization of the tumor vasculature as well as starving the tumor, enhance the delivery within the tumor of anticancer agents and make the microenvironment favorable to immunotherapy.

## 5. Overcoming Resistance: Depicting the Future of HCC Treatment 

Progress in understanding mechanisms of resistance to anti-angiogenetic drugs has led to the discovery of new therapeutic strategies for HCC patients, resulting in advances that will soon reshape the current clinical scenario in metastatic disease (Table 1).

### 5.1. Combination Strategies with Approved Antiangiogenetic Drugs

#### 5.1.1. Sorafenib

Several drugs have been studied in combination with sorafenib in order to revert (or prevent) resistance to it.

MEK inhibition, such as cobimetinib, was shown to inhibit the MAPK pathway in preclinical models of HCC, improving the inhibitory effects of sorafenib [93]. However, a phase I trial in treatment-naïve HCC patients treated with the combination of sorafenib and trametinib, another MEK inhibitor, did not register objective responses [94]. Galunisertib is an antagonist of the tyrosine kinase TGF-β receptor type 1 (TGFBR1), which is involved in angiogenesis during HCC development [95]. A combination of sorafenib and galunisertib as first-line treatment was investigated in a phase II trial demonstrating clinical feasibility, even if patients without tumor response had worse survival, thus highlighting the need for appropriate biomarker(s) in patient selection [96].

Immunotherapy could potentiate sorafenib efficacy and anti-PD-1 antibodies are administered in combination with it in ongoing clinical trials (NCT03439891, NCT03211416, NCT02988440); however, the already published results of the IMbrave150 trial [97] have raised the bar of combining immunotherapeutic and antiangiogenic drugs, questioning the optimal drug association as a first-line approach.

New promising drugs are currently under assessment in translational projects. Torin-2 is an ATP-competitive dual mTORC1/2 inhibitor tested in preclinical sorafenib-resistant HCC models; it showed a synergistic effect with sorafenib through suppression of the PI3K/AKT/mTOR pathway [98]. Acting on the same pathway, valproic acid sensitizes resistant HCC cells to sorafenib in preclinical experiments [99]; also, the Notch1 pathway is involved in valproic acid action in sorafenib-resistant HCC cells, underlying the great potentiality of this molecule in this specific setting [100]. Specific drugs, such as miransertib which synergistically inhibits tumor progression with sorafenib [101], could inhibit AKT.

CKD-5 is a pan-histone deacetylase inhibitor (HDACI) which has been shown to be more effective in inhibiting, together with sorafenib, HCC cells compared to panobinostat, which is an HDACI already approved for multiple myeloma treatment [102]. Ibrutinib, a TKI of BTK and ErbB family members, by inactivating downstream Akt and MAPK pathways, is also a potential partner for sorafenib in treating HCC by modulating immune cells in the stroma through BTK inhibition [103].

#### 5.1.2. Lenvatinib

The few data on combination therapy with lenvatinib available mostly stemmed from preclinical studies in which lenvatinib was associated with natural compounds as follows: chelidonine, extracted from Chelidonium majus L., reverts lenvatinib resistance through inhibition of EMT [104]; sophoridine, extracted from the seeds of Sophora alopecuroides L., inhibits MAPK pathway in lenvatinib-resistant cells [105]; oxysophocarpine, extracted from Siphocampylus verticillatus, sensitizes HCC cells overexpressing FGFR1 to lenvatinib [106]. Recently, the phase III LEAP-002 study (NCT03713593) investigated lenvatinib in combination with pembrolizumab as the first-line treatment of patients with unresectable HCC [107].

Lenvatinib is also under investigation in combination with toripalimab, an anti-PD1 antibody (NCT04523493) and in a phase II trial with a co-formulation of pembrolizumab and quavonlimab, an anti-CTLA4 antibody (NCT04740307).

#### 5.1.3. Regorafenib and Cabozantinib

Very little preclinical data about reverting resistance to regorafenib and cabozantinib have been published to date. Of interest, recent work suggested that combining cabozantinib with MLN0128, an mTOR inhibitor, could obtain HCC regression in both in vitro and in vivo models [108].

Regarding association with immunotherapy, both regorafenib and cabozantinib, which are currently approved as second-line therapies, are now being tested as first-line approaches in treatment-naïve HCC patients. In particular, regorafenib has been tested in combination with anti-PD1 antibody tislelizumab (NCT04183088, phase II trial) or with nivolumab (NCT04310709, phase II trial) and cabozantinib with atezolizumab (NCT03755791, phase III trial) or pembrolizumab (NCT04442581, phase II trial).

### 5.2. Experimental Single-Agent Drugs

#### 5.2.1. Antiangiogenetic Drugs

Several new molecules with different kinase inhibition profiles are under investigation in HCC patients.

Apatinib is an oral VEGFR-2 (Flk-1/KDR), RET, c-Kit and c-Src inhibitor, which can also revert multidrug resistance by inhibiting ATP-binding cassette transporters [79]. It has been tested in sorafenib-resistant HCC patients with an improved OS compared to best supportive care (BSC), even if a longer OS was observed among patients with low tumor burden [80].

Anlotinib is an oral antiangiogenetic TKI, which, unlike the other already approved drugs, inhibits all VEGFR (1, 2 and 3) and FGFR (1, 2, 3 and 4) isoforms and both PDGFR subunits [81]; an ongoing phase II trial will clarify its clinical role after lenvatinib in HCC patients (NCT04080154).

Donafenib is an oral inhibitor of RAF and VEGFR, which has been shown to prolong OS over sorafenib as first-line therapy, even with a similar PFS, in a phase II/III trial [82]. Despite being better tolerated than sorafenib, it would be difficult for it to become an option as a first-line approach in light of IMbrave150 trial results.

Sitravatinib is an oral antiangiogenetic TKI, which displays an inhibition profile similar to cabozantinib, since it blocks VEGFR, PDGFR (α,β), c-Kit, MET and AXL, but also members of the Ephrin receptor family (i.e., EphA1). The broad-spectrum inhibition of sitravatinib could determine a greater suppression of parallel pathways involved in molecular resistance to other antiangiogenetic drugs, thus explaining interest in this molecule [83]. A phase I/II trial is currently evaluating the role of sitravatinib alone or in combination with tislelizumab (anti-PD1 antibody) in HCC patients (NCT03941873).

Among FGFR inhibitors, infigratinib, which is a pan-FGFR inhibitor, showed potent antitumor activity in HCC preclinical models [84], and two recent studies highlighted the efficacy of this molecule in combination with bevacizumab [109] and the CDK4/6 inhibitor ribociclib [110]; to date, no clinical trial has investigated this molecule in HCC patients. FGFR4 has been indicated as a better candidate for inhibition in HCC preclinical models and new specific inhibitors are under development [85].

#### 5.2.2. Other Molecules

MET receptor and its ligand, HGF, are overexpressed in a good percentage of HCC patients [86]. However, tivantinib, an oral MET inhibitor, failed to improve OS in pretreated MET-high HCC patients in two phase III clinical trials [87]; these disappointing results suggest that other mechanisms are responsible for tumor cells’ survival in this specific subset of HCC patients, requiring more efforts in identifying them. 

Nearly two-thirds of HCC patients express epithelial growth factor receptor (EGFR) family members [88]. Varlitinib is an oral reversible pan-HER inhibitor, currently being tested in a phase Ib trial (NCT03499626) in patients who progressed on first-line sorafenib or lenvatinib, after interesting results were obtained in HCC xenograft models [89]. 

Inflammation could drive HCC progression, and among molecular mechanisms, JAK/STAT3 pathway controls survival, neoangiogenesis, immune surveillance and metastasizing of HCC [110]. Itacitinib is an oral selective inhibitor of JAK1 [90], which is now being tested in pretreated HCC patients in the phase Ib “JAKaL” trial (NCT04358185). Acting on the same pathway, but also modulating NK and CD8+ T cells activity, a new oral drug called icaritin has already been shown to be a potential immunotherapy agent in advanced HCC [91]. It is now being tested versus sorafenib in a phase III trial (NCT03236649) in treatment-naïve PD-L1 + HCC patients, in light of promising results from a phase II trial [92].

### 5.3. Combination of Immunotherapy and Antiangiogenic Drugs

#### 5.3.1. IMbrave 150 Trial: Pioneer of Successful Combination Therapy in HCC

In order to overcome resistance to common targeted therapy, the advent of immunotherapy represented a new opportunity while raising, however, a new problem: resistance to the ICIs themselves.

Currently, immunotherapy with the combo ICI and anti-angiogenetic drug proved for the first time to be the most effective systemic treatment for advanced, unresectable HCC, in the phase III Imbrave 150 trial [111,112].

Based on these results, further investigations concerning the combo were carried out (Figure 3) [113,114,115,116,117]. 

#### 5.3.2. Immunotherapy and Heterogeneous Responses: Role of the Immune Landscape in the Tumor Microenvironment

The encouraging results from the IMbrave150 trial [11] led to various combination strategies to offer improved efficacy and to help overcome the condition of both primary and acquired resistance to immunotherapy. In fact, only approximately 20% of advanced HCC patients benefit from ICIs and most of them have disease progression after 3–9 months [118]. 

Heterogeneity in the immune microenvironment contributes the different response rates to ICIs [119]. Based on the characteristics of the TME, and particularly on the presence of tumor-infiltrating CD8 + T cells and the expression of PD-L1, patients with HCC could be stratified into two clusters—the responsive cluster and the resistant cluster—or in four types, accordingly [120]. The responsive cluster consists of the immune hot subclass and the immune moderate subclass. Tumors belonging to the immune hot subclass (type I tumors, about 30% of HCCs), contain tumor-infiltrating lymphocytes (TILS) and express PD-L1 [121]. Type I tumors generally show adequate response to PD-1/PD-L1 antibodies monotherapy. In type I, there is an initial antitumor immune response, the cancer cells escape the activated CD8 + T cells, and they carry out the immune escape by upregulating the expression of PD-L1. This is why type I tumors are responsive to monotherapy with anti-PD-L1 antibodies. Regarding gene signatures, mutations and chromosomal aberrations, this group is usually characterized by the activation of the JAK/STAT3 pathway resulting in increased expression of PD-L1 and MHC molecules and by Wnt/Tgf-B canonical interacting pathways; there are no chromosomal aberrations. 

Tumors belonging to the immune moderate subclass (type IV tumors, about 40% HCCs) lack PD-L1 expression although they contain TILs; type IV tumors do not respond to monotherapy with anti-PD-L1 antibodies because the immunosuppressive TME inhibits CD8 + cells. These tumors are never attacked by CD8+ cells. Therefore, there is no immune escape neither is there PD-L1 overexpression. Therefore, ICIs are not effective even if there are a large number of TILs. In such tumors, the anti-VEGFs can reprogram the immunosuppressive microenvironment into an immunostimulating microenvironment and the concomitant therapy with PD1/PDL1 antibodies and VEGF antibodies or TKI could be effective.

Type II and III tumors (about 30% HCCs) represent the resistant cluster or immune cold subclass. Type II tumors lack the presence of TILs and PDL1 expression; they are characterized by immunological ignorance. This group needs to enhance immunogenicity by increasing the infiltration of T cells in the tumor (i.e., by adoptive cell therapy). From a point of view of gene signatures and chromosomal aberrations, both mutations in the Wnt/ β-catenin pathway and chromosomal aberrations can be found. In general, Wnt/β-catenin plays an important role in the epithelial-to-mesenchymal transition (EMT) process in the TME [122]. Wnt/β-catenin activation mutations are known to contribute to primary resistance to ICIs and are present in approximately 40% of HCCs, especially in well-to-moderately differentiated HCCs that are from the Jekyll phenotype (non-progressive) to Hyde phenotype (progressive). HNF4α, a target gene of Wnt, suppresses EMT, resulting in a non-aggressive phenotype with less metastasis and invasion; high HNF4α status characterizes the Jekyll phenotype [123]. FOXM1 instead, which is regulated by Wnt, promotes EMT and decreases E-cadherin expression, resulting in a poor prognostic phenotype with aggressive metastasis and invasion [124]. Type III tumors are defined by both the absence of TILs and the expression of PD-L1. They need to increase immunogenicity [125], for example through oncogenic pathway induction of PD-L1 (oncolytic viruses). These subclasses were designed using samples from tumors that have never been exposed to ICI and therefore, they could only be attractive hypotheses. However, recent literature [126] has identified, from pre- and post-treatment paired biopsies, key molecular correlates of response and resistance to atezolizumab, an anti-PD-L1 antibody, in combination with the anti-VEGF bevacizumab confirming that anti -VEGF synergizes with anti-PD-L1 targeting angiogenesis, Treg proliferation and myeloid cell inflammation.

#### 5.3.3. Combining Immunotherapy in First-Line Setting: Recent Clinical Trials

We summarize in Table 2 the data from first-line ICI combination phase III studies, which have a high chance of bringing a profound change to the HCC clinical management as their primary endpoints have already been met.

HIMALAYA trial (NCT03298451) results were recently shown at the 2022 ASCO GI Cancer Symposium [127]; SHR-1210-III-310 trial (NCT03764293) results were recently shown at the 2022 ESMO Congress [128] (Table 2).

## 6. Changing the Algorithm to Optimize the “Continuum of Care” and Future Perspectives

One of the big issues about advanced HCC patients is drafting the optimal antiangiogenetic drug algorithm in order to tailor the treatments and define a “continuum of care” for these patients [129]. It seems unlikely that all patients can benefit from the same therapeutic sequence, but to date, clinical practice has been bound to specific inclusion criteria of clinical studies. 

Sorafenib, which has been the first-line standard of treatment for many years, was first resized by the approval of lenvatinib and has now been overtaken by the combination of atezolizumab and bevacizumab. However, in the second-line setting, all drugs have been tested in a sorafenib-resistant population, thus precluding their use in patients treated with the new drugs; for the latter, sorafenib remains the only available therapeutic option but there are various aspects that make the scenario even more complex. First of all, anti-angiogenic therapies are limited by drug resistance [42,43]. For this reason, several combinations of approved targeted therapies in HCC and other partner drugs are being investigated. These are inhibitors targeting pathways other than the VEGFR or c-met, TGFB, mTORC1/2, AKT and FGFR1. Other clinical trials are now investigating the use in monotherapy of different antiangiogenetic drugs or other multi-kinase inhibitory molecules.

Immunotherapy, on the other hand, which might have represented a way of overcoming this issue, has highlighted other problems such as heterogeneous responses and also resistance, partially overcome by the combined treatment of conventional targeted therapy and ICIs [130].

In addition, the normalization of the vasculature is an emerging concept that has its goal in the enhancement of tumor perfusion and therefore oxygenation in order to promote the delivery and effectiveness of the therapies [72]. Preclinical studies showed that low doses of anti-VEGFR2 antibodies during the window of normalization might lead to an increase in tumor infiltration by T cells, especially if the antiangiogenetic treatment is carried out several days before an ICI. Therefore, having established the role of antiangiogenic in reprogramming the tumor microenvironment from an immunosuppressive to an immuno-supportive one, the concept of vascular normalization perhaps has led us to more critically evaluate the combination with ICIs, i.e., the dose, the duration and the schedule of administration of the antiangiogenic (e.g., simultaneous or sequential) (Figure 4) [131,132,133].

ANG2 could confer resistance to anti-VEGF therapy, whereas targeting both VEGF and ANG2 could increase vasculature normalization and promote antitumor immunity [134]. Several preclinical studies have investigated the dual VEGF-ANG2 blockade in different cancers using a bispecific anti-VEGF-ANG2 antibody. However, the dual VEGF-ANG2 blockade alone does not allow clinical responses even if it has stressed the importance of dosing and scheduling. In fact, dual angiogenetic treatment could cause excessive pruning of vessels and prevent the delivery of drugs to the tumor [135]. Of interest, ANG2 levels decrease during the normalization window, so this aspect could be useful for defining the appropriate time to add the immunotherapy. Anyhow, ANG2 needs to be explored as a potential predictive and/or prognostic biomarker [136] because high levels of ANG2 in the tumor or in the circulation are correlated with unfavorable patient survival in different cancers including HCC, possibly reflecting hypoxia and immunosuppression in the TME.

Another key aspect is regarding the Wnt/β-catenin pathway, since activating mutations are known to contribute to primary resistance to ICIs. A recent publication defined that Wnt/β-catenin mutations in HCCs have a dual phenotype: the Jekyll and Hyde phenotype (Figure 4) [137]. 

The Jekyll phenotype, characterized by good differentiation and less vascular invasiveness, is strongly associated with high expression of HNF4α. HNF4α promotes the expression of OATP1B3, a transporter of bile acids, which determines a higher enhancement of intrahepatic nodules in the hepatobiliary phase of Gd-EOB-DTPA-enhanced magnetic resonance imaging, commonly used for HCC diagnosis. The Hyde phenotype—with poor differentiation and massive vascular invasion—is strongly associated with high expression of FOXM1; this transcription factor promotes the expression of GLUT1 and may be recognized as a hot nodule in FDG-PET/CT images. A possible explanation is that the first should be mainly resistant to ICIs, while the last one is immune-activated and exhausted. However, further future non-invasive applications should be developed in this field.

## 7. Conclusions

Drug sequencing in HCC needs more data on both the clinical and translational sides. In the era of precision oncology, it would be reasonable to better characterize each tumor from the molecular point of view in order to identify potential resistance mechanisms and guide the treatment choice accordingly. Besides this, in the context of immunotherapy, it is important to define, in the combination strategies, the most appropriate administration, if simultaneous or sequential.

## Figures and Tables

**Figure 1 cancers-14-06245-f001:**
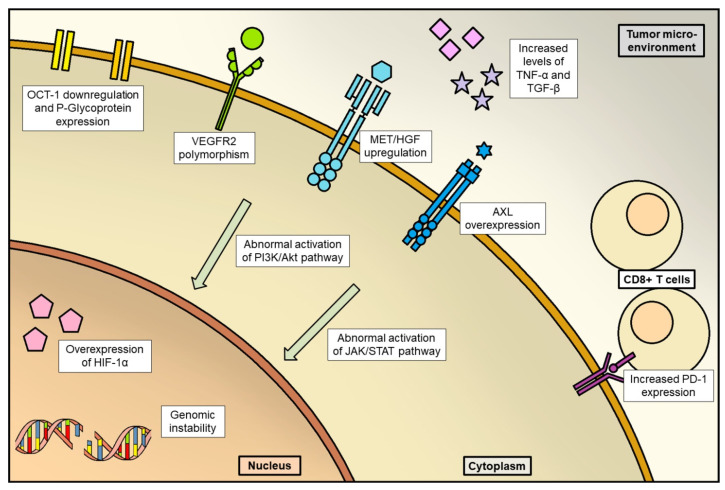
Main mechanisms of anti-angiogenic drug resistance in HCC. In this figure, the main mechanisms of resistance are summarized, represented at three levels: intranuclear, intracytoplasmic and tumor microenvironment.

**Figure 2 cancers-14-06245-f002:**
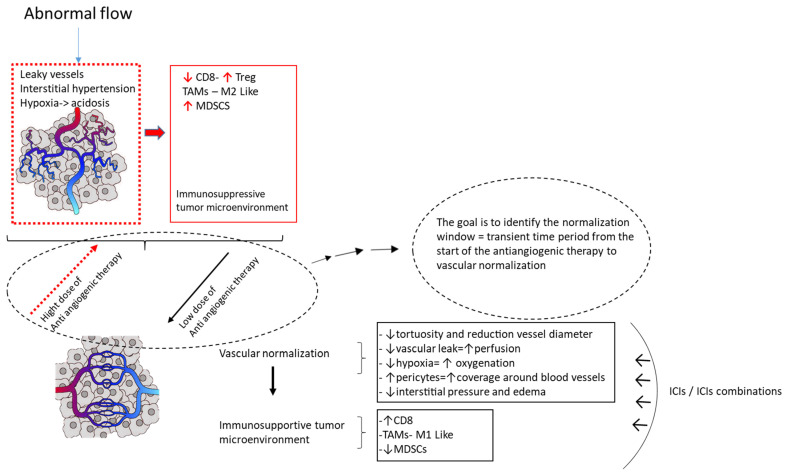
Liver tumor display vascular abnormalities. Liver tumor vessels have abnormal blood flow and are excessively leaky. This results in hypoxia and acidosis which contribute to immunosuppression in the TME: expansion of immunosuppressive regulatory T (Treg) cells and of myeloid-derived suppressor cells (MDSC); decrease in the infiltration of the CD8+ T cells; reprogramming of tumor-associated macrophages (TAMs) from an anticancer M1-like phenotype towards the pro-tumor M2 phenotype. The normalization process is transient and its onset provides an immune-supportive microenvironment, an efficient infiltration of immune cells and the delivery of anticancer therapy including immunotherapy. A judicious application of antiangiogenic therapy, neither too high nor too low, allows for the normalization window and the related benefits to be obtained.

**Figure 3 cancers-14-06245-f003:**
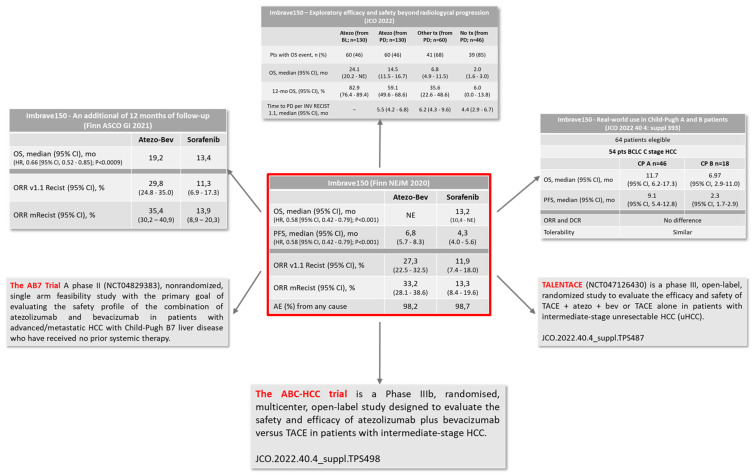
The IMbrave150 trial and further updates.

**Figure 4 cancers-14-06245-f004:**
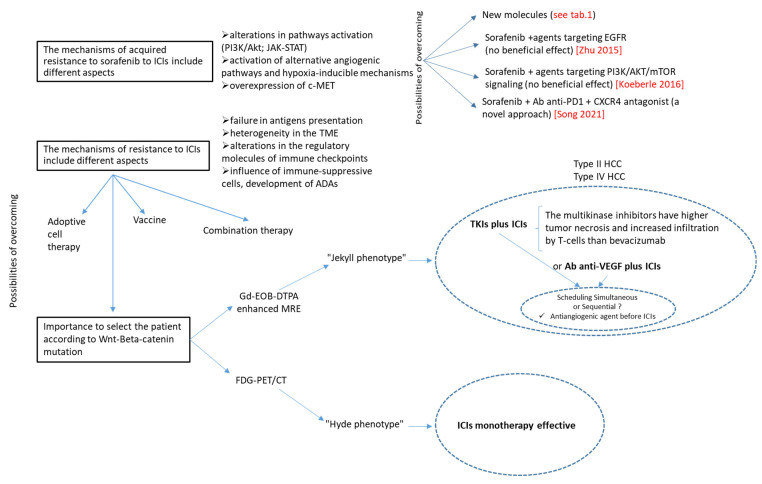
How to overcome the resistance: an overview. This schematic representation underlines the open questions about resistance to antiangiogenic therapy and immunotherapy and the possibilities of overcoming them as well as the importance of stratifying patients into different subtypes to achieve higher response rates. Abbreviations: c-MET, c-mesenchymal-epithelial transition receptor; PI3K/Akt, phosphatidylinositol-3-kinase/Akt; mTOR, mammalian target of rapamycin; ADAs, anti-drug antibodies; EGFR, epidermal growth factor receptor; PD-1, programmed cell death; CXCR4, CXC chemokine receptor type 4; TKI, tyrosine kinase inhibitor; ICIs, immune checkpoint inhibitors; Gd-EOB-DTPA enhanced MRE, gadolinium-ethoxybenzyl-diethylenetriamine-enhanced magnetic resonance; FDG-PET/CT, fluoro-2-deoxy-D-glucose-PET/CT.

**Table 1 cancers-14-06245-t001:** New molecules overcoming resistance to approved anti-angiogenetic drugs in both preclinical and clinical settings.

Molecule	Mechanism of Action	Evidence	Reference
Antiangiogenetic drugs			
Apatinib	VEGFR-2, RET, cKIT and cSRC inhibitor	Phase II/III trial (post- sorafenib patients): improved OS	Mi et al. [79] Zhang et al. [80]
Anlotinib	VEGFR(1,2,3), FGFR(1,2,3,4), PDGFR(α,β) and cKIT inhibitor	Phase II trial (post-lenvatinib patients): ongoing	Shen et al. [81]
Donafenib	RAF and VEGFR inhibitor	Phase II/III trial (versus sorafenib): improved OS	Bi et al. [82]
Sitravatinib	VEGFR, PDGFR(α,β), cKIT, MET, AXL and EphA1 inhibitor	Preclinical studies	Dolan et al. [83]
Infigratinib	Pan-FGFR inhibitor	Preclinical studies (alone or in combination)	Huynh et al. [84] Rezende et al. [85]
Other molecules			
Tivantinib	MET inhibitor	Phase III trials (post-sorafenib MET-high patients): negative in OS	Rimassa et al. [86] Kudo [87]
Varlitinib	Pan-HER inhibitor	Phase Ib trial (post-sorafenib or post-lenvatinib patients): ongoing	Ito et al. [88] Hsieh et al. [89]
Itacitinib	JAK1 inhibitor	Phase Ib trial (post-sorafenib or post-lenvatinib patients): ongoing	Covington et al. [90]
Icaritin	JAK2/STAT3 inhibitor, immune modulator	Phase III trial (versus sorafenib in PD-L1+ patients): ongoing	Fan et al. [91] Sun et al. [92]

Abbreviations: VEGFR: vascular endothelial growth factor receptor; RET: Rearranged during Transfection; cKIT: proto-oncogene tyrosine-protein kinase KIT (known as CD117); cSRC: proto-oncogene tyrosine-protein kinase SRC; FGFR: fibroblast growth factor receptor; RAF: RAF proto-oncogene serine/threonine-protein kinase; MET: tyrosine-protein kinase MET; AXL: tyrosine-protein kinase receptor UFO; PDGFR: platelet-derived growth factor receptor; EphA1: ephrin type-A receptor 1; PD-L1: programmed death ligand 1; HER: human epidermal growth factor receptor; JAK1: Janus kinase 1; JAK2/STAT3: Janus kinase 2/signal transducer and activator of transcription 3; OS: overall survival.

**Table 2 cancers-14-06245-t002:** First-line ICI combination phase III studies.

	Median OS, mo (95% CI)	Median PFS, mo (95% CI)	ORR
Himalaya (NCT03298451)			
Primary objective: OS for STRIDE vs. S Secondary objective: non-inferiority for OS for D vs. S			
STRIDE (*n* = 393) Durvalumab (*n* = 389) Sorafenib (*n* = 389)	16.4 (14.2–19.6) 16.6 (14.1–19.1) 13.8 (12.3–16.1)	3.8 (3.7–5.3) 3.7 (3.2–3.8) 4.1 (3.8–5.5)	20.1% 17.0% 5.1%
SHR-1210-III-310 (NCT03764293)			
Primary objective: OS and PFS for C + R vs. S Secondary objective: ORR			
Camrelizumab + Apatinib Sorafenib	22.1 (19.1–27.2) 15.2 (13.0–18.5)	5.6 (5.5–6.3) 3.7 (2.8–3.7)	25.4% 5.9%

Abbreviations: OS: overall survival, PFS: progression-free survival; ORR: overall response rate; S: sorafenib; D: durvalumab; C + R: camrelizumab + apatinib; STRIDE: single tremelimumab with regular interval of durvalumab.

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
