# Peer review of "Resistance to Antiangiogenic Therapy in Hepatocellular Carcinoma: From Molecular Mechanisms to Clinical Impact"

_cancers, 2022, doi:10.3390/cancers14246245_

Round 1
Reviewer 1 Report
Line 65: immunotherapy a way to overcome resistance to anti-angiogenic agents – isn’t it considered to be the other way around? This is also what the authors write in line 237
Line 68: 20% of patients does not respond to ICI – in fact, this percentage is much higher
Line 171: to be non-inferior is not the same as “to be as effective as”.
Line 338: results of LEAP002 could be mentioned
Section 5.3.2 contains a lot of strong statements about the relation between molecular subclasses of HCC and response to ICI (with or without VEGFi). The truth is, a lot is speculative and most of the subclasses were developed using samples from tumors that were never exposed to ICI. So, while these are attractive hypotheses, they remain hypotheses. I would recommend to include the Zhu et al Nat Med 2022 paper that has information on molecular level associated with response to atezolizumab and atezolizumab/bevacizumab.
Section 5.3.3 is about ongoing trials but only mentions already finished trials with reported data
Author Response
Dear Reviewer,
thank you for your support. Below are the responses to your comments and suggestions:
Line 65: immunotherapy a way to overcome resistance to anti-angiogenic agents – isn’t it considered to be the other way around? This is also what the authors write in line 237
Line 65: thank you for your comment; we have rephrased it in order to avoid misunderstandings:
“Thus, if immunotherapy might be considered a way to improve the efficacy of targeted therapies,…”
Line 68: 20% of patients does not respond to ICI – in fact, this percentage is much higher
Line 68: thank you for your observation, we corrected with a new reference:
“in fact, 40% of patients are primarily resistant to immune checkpoint inhibitors (ICIs) [De Lorenzo, S. Cancers. 2022 Sep 23]”.
Line 171: to be non-inferior is not the same as “to be as effective as”.
Line 171: thank you for your observation, we have corrected by replacing the sentence with the following one: “Lenvatinib has shown to be non –inferior to sorafenib in prolonging OS”
Line 338: results of LEAP002 could be mentioned
Line 338: thank you for your suggestion; we corrected mentioning the study LEAP002 and adding the reference: “Recently the phase 3 LEAP-002 study (NCT03713593) investigated lenvatinib in combination with pembrolizumab as first-line treatment of patients with unresectable HCC” [Finn, R.S. Ann Oncol. 2022].
Section 5.3.2: contains a lot of strong statements about the relation between molecular subclasses of HCC and response to ICI (with or without VEGFi). The truth is, a lot is speculative and most of the subclasses were developed using samples from tumors that were never exposed to ICI. So, while these are attractive hypotheses, they remain hypotheses. I would recommend to include the Zhu et al Nat Med 2022 paper that has information on molecular level associated with response to atezolizumab and atezolizumab/bevacizumab.
Section 5.3.2: thank you for your observation and suggestion; we corrected by including the mentioned paper: “These subclasses were designed using samples from tumors that have never been exposed to ICI and therefore, they could only be attractive hypotheses. However, recent literature [Zhu AX. Nat Med. 2022] has identified, from pre- and post-treatment paired biopsies, key molecular correlates of response and resistance to atezolizumab, an anti-PD-L1 antibody, in combination with the anti-VEGF bevacizumab confirming that anti -VEGF synergizes with anti-PD-L1 targeting angiogenesis, Treg proliferation and myeloid cell inflammation” .
Section 5.3.3: is about ongoing trials but only mentions already finished trials with reported data
Section 5.3.3: Thanks for your observation, we modified the title accordingly and modified the table.
Date of the re-submission
08 Dec 2022

Reviewer 2 Report
Resistance to antiangiogenic therapy provides difficulty in the treatment of late-stage HCC. The authors evaluated antiangiogenic medication resistance, concentrating on the tumor microenvironment and vascular normalization, in order to provide potential overcoming strategies. It is an intriguing study for the present treatment of HCC patients with antiangiogenic medication resistance.
My comments:
1. The authors concentrated on hypoxia and the immune milieu in the tumor microenvironment of drug resistance. They did not, however, consider the consequences of viral reactivation. Because a large number of HCC patients have a history of hepatitis B/C virus infection, I suggest the authors supplement the relevant information.
2. Some references, such as the Introduction section (page 2, paragraph 3), were omitted.
Author Response
Dear Reviewer,
thank you for your support. Below are the responses to your comments and suggestions:
Resistance to antiangiogenic therapy provides difficulty in the treatment of late-stage HCC. The authors evaluated antiangiogenic medication resistance, concentrating on the tumor microenvironment and vascular normalization, in order to provide potential overcoming strategies. It is an intriguing study for the present treatment of HCC patients with antiangiogenic medication resistance.
1.The authors concentrated on hypoxia and the immune milieu in the tumor microenvironment of drug resistance. They did not, however, consider the consequences of viral reactivation. Because a large number of HCC patients have a history of hepatitis B/C virus infection, I suggest the authors supplement the relevant information.
1. Thank you for your observation. We added relevant information about HBV and HCV infections on HCC genesis. “Another aspect that should be considered is the active role of hepatitis B virus (HBV) and hepatitis C virus (HCV) in carcinogenesis and tumor progression in HCC [Elpek2021]. In detail, HBV DNA integrates in host chromosomes, causing gene de-regulation in preferred target regions, but also on the epigenetic level through DNA methylation and/or histone modifications: all these alterations cause impairment of the main pathways involved in HCC development, including Wnt/β-Catenin and angiogenic ones [Daud2017][Tang2005]. Concerning HCV, a similar influence on molecular pathway has been highlighted [Liu2011][Shao2017].
2. Some references, such as the Introduction section (page 2, paragraph 3), were omitted.
2. Thank you for this comment, we added omitted references (Goel et al 2011, Magnussen et al 2021, De Lorenzo et al 2022). We also added some missing references in the chapter 3 (Al-Abd et al 2017, Wang et al 2019, Asic 2015).
Date of the re-submission
08 Dec 2022

Reviewer 3 Report
Federico et al. compose a comprehensive look into the mechanisms of drug resistance regarding the standard of care for HCC. This review is timely, insightful and informative to the readership. Other than minor text revisions required, this work is exciting and is a unique look regarding the anti-angiogenic side of the bev+atezo combination that is quickly becoming the standard of care in the clinic.
Author Response
Comments and Suggestions for Authors
Federico et al. compose a comprehensive look into the mechanisms of drug resistance regarding the standard of care for HCC. This review is timely, insightful and informative to the readership. Other than minor text revisions required, this work is exciting and is a unique look regarding the anti-angiogenic side of the bev+atezo combination that is quickly becoming the standard of care in the clinic.
Dear Reviewer, thank you for your support. We have provided to required revisions.
Date of the re-submission
08 Dec 2022

Reviewer 4 Report
The present review summarized the challenges of resistance to antiangiogenic therapy in hepatocellular carcinoma (HCC) treatment and the potential mechanisms. Both molecular mechanisms and clinical strategies were proposed. The review is overall comprehensive and insightful. Some specific points are listed as below.
1. The molecular mechanisms underlying the resistance to antiangiogenic therapy was not extensively introduced and discussed. A figure to summarize the proposed models should be presented.
2. The subtitles of the review could be better optimized.
Author Response
Dear Reviewer,
thank you for your support. Below are the responses to your comments and suggestions:
The present review summarized the challenges of resistance to antiangiogenic therapy in hepatocellular carcinoma (HCC) treatment and the potential mechanisms. Both molecular mechanisms and clinical strategies were proposed. The review is overall comprehensive and insightful. Some specific points are listed as below.
1. The molecular mechanisms underlying the resistance to antiangiogenic therapy was not extensively introduced and discussed. A figure to summarize the proposed models should be presented.
1. Thank you for your observation. We added “Figure 1. Main mechanisms of anti-angiogenic drug resistance in HCC. In this figure are summarized the main mechanisms of resistance, represented at three levels: intranuclear, intracytoplasmic and in tumor microenvironment”.
2. The subtitles of the review could be better optimized.
2. Thank you for your observation. We have provided.
Date of the re-submission
08 Dec 2022
